# Gen-LRA: Towards a Principled Membership Inference Attack for Generative Models

## Abstract

Evaluating the potential privacy leakage of synthetic data is an important but un-resolved problem. Most existing adversarial auditing frameworks for synthetic data rely on heuristics and unreasonable assumptions to attack the failure modes of generative models, exhibiting limited capability to describe and detect the privacy exposure of training data. In this paper, we study designing Membership Inference Attacks (MIAs) that specifically exploit the observation that generative models tend to memorize certain data points in their training sets, leading to significant local overfitting. Here, we propose Generative Likelihood Ratio Attack (Gen-LRA), a novel, computationally efficient shadow-box MIA that, with no assumption of model knowledge or access, attacks the generated synthetic dataset by conducting a hypothesis test that it is locally overfit to potential training data. Assessed over a comprehensive benchmark spanning diverse datasets, model architectures, and attack parameters, we find that Gen-LRA consistently dominates other MIAs for generative models across multiple performance metrics. These results underscore Gen-LRA's effectiveness as an interpretable and robust privacy auditing tool, highlighting the significant privacy risks posed by generative model overfitting in real-world applications.

## 1 Introduction

Real world tabular data is often privacy-sensitive to the individual observations that compose these samples, hindering their ability to be shared in open-science efforts that can aid in new research and improve reproducibility. A promise of generative modeling is that models trained on private data can produce samples that preserve the privacy of the training set while maintaining much of the original statistical information. In practice, a wide array of methodologies have been proposed to accomplish this involving modifying loss functions (Abadi et al., 2016; Wang et al., 2022), creating new architectures (Yoon et al., 2019; 2020a), and studying data release strategies (Hardt et al., 2012; Gupta et al., 2012; Takagi et al., 2021) to provide a guarantee of differential privacy. In another direction, many methods have been proposed that maximize the fidelity of synthetic data and argue they are private through similarity metrics like average Distance to Closest Record that evaluate overfittness (Zhao et al., 2021; Guillaudeux et al., 2022; Liu et al., 2023).

While both lines of synthetic data research (private and non-private) have seen rapid advancements, techniques to evaluate the empirical privacy of these generative models have lagged behind. Auditing differentially private algorithms can be methodologically challenging (Jagielski et al., 2020; Chua et al., 2024) and from a practitioner perspective, theoretical notions of privacy can be difficult to practically interpret. For non-differentially-private models, similarity metrics between the training and synthetic sets have been argued to be heuristic as they do not actually characterize privacy risk but rather an ad-hoc measure of overfitting (Platzer & Reutterer, 2021; Ganev & Cristofaro, 2023; Ward et al., 2024).

Recently, Membership Inference Attacks (MIAs) have shown to be a computationally efficient, powerful, and interpretable framework for evaluating the empirical privacy of machine learning models by attacking overfitting (Shokri et al., 2017; Chen et al., 2020; Carlini et al., 2021). Here, privacy auditing is posed as a game where an adversary, given a threat model that describes what information can be used, constructs an attack that classifies whether a test observation is a member of the dataset a model was trained with. A successful attack represents a practical and interpretable pri-

vacy breach. As a classic example, an insurance company could have access to a hospital's synthetic cancer dataset and, for a new applicant, attack the dataset to determine if the applicant is a member, leaking their diagnosis (Hu et al., 2022).

While promising, MIAs for generative models and synthetic data release have seen limited success. Previous work in Generative Model MIAs often relied on heuristics for the attack and usually explored including additional assumptions about model access that have been argued to be unrealistic (van Breugel et al., 2023; Ward et al., 2024). In contrast, we focus on studying Membership Inference for synthetic data release in a shadow-box threat model (Chen et al., 2020) where we make minimal assumptions about model architecture, model access and model quality in deriving a powerful MIA called Generative Likelihood Ratio Attack (Gen-LRA) which utilizes a hypothesis testing framework to target privacy leakage from model overfitting. We show that our attack broadly outperforms competing methods especially at low fixed false positive rates, highlighting that overfitting presents a more dangerous source of privacy leakage then previously suggested, even in differentially private generative models. Our contributions are as follows:

**Contributions**:

1. We introduce Gen-LRA, a novel MIA that uses Likelihood Ratio framework to attack overfitting in generative models with minimal assumptions by evaluating the likelihood of Synthetic data under a null and alternative hypothesis that the model is overfit to a potential training example.

2. We show that Gen-LRA is computationally efficient and broadly outperforms other MIAs for generative models across a diverse benchmark of datasets, model architectures, and attack parameters.

3. We demonstrate that Gen-LRA identifies a different source of privacy leakage relative to other commonly used MIAs. Worryingly, we also show that this privacy leakage can occur non-randomly relative to different sub-groups that compose the training data. This indicates that even if a model is robust to MIAs in the aggregate, it can still leak the data of outlier data points in the training set.

## 2 MEMBERSHIP INFERENCE ATTACKS FORMALISM

In this work, we specifically study the Membership Inference Attack Game in the context of *synthetic data generation*. The objective of this game is to determine whether a particular data point was included in the original training dataset by examining the outputs of a generative model. We first introduce the formal definition of the *Membership Inference Attack Game*:

**Definition (Membership Inference Attack Game).** The game proceeds between a challenger $\mathcal{C}$ and an adversary $\mathcal{A}$ as follows:

1. The challenger samples a training dataset $T = x_i{}_{i=1}^n$ from the population distribution $x_i \sim \mathbb{P}$ and uses $T$ to train a tabular generative model $G \leftarrow \mathcal{T}(T)$. The generative model $G$ produces synthetic dataset $S$.

2. The challenger flips a bit $b \in 0, 1$. If $b = 0$, the challenger samples a test observation $x^\star$ from the population distribution $\mathbb{P}$. Otherwise, the challenger selects the test observation $x^\star$ from the training set $T$.

3. The challenger sends the test observation $x^\star$ to the adversary $\mathcal{A}$.

4. The adversary has access to some information defined by a threat model and uses this information to output a guess $\hat{b} \leftarrow \mathcal{A}(x^\star)$.

5. The output of the game is 1 if $\hat{b} = b$, and 0 otherwise. The adversary wins if $\hat{b} = b$, i.e., if it correctly identifies whether the test observation $x^\star$ was part of the training set $T$ or a freshly sampled data point from the population distribution $\mathbb{P}$.

**Adversary's Goal and Capabilities**   The adversary $\mathcal{A}$ in the Membership Inference Game aims to determine whether a specific data point $x^\star$ was part of the original training dataset $T$ or was drawn from the population distribution $\mathbb{P}$. Here, the adversary can utilize available information in any

manner to construct a method to classify the membership of $x^*$. The performance of the classifier, which can be evaluated with binary classification metrics, is a measure of the privacy leakage of the training data from $G$ and $S$. Formally, this classification or Membership Inference Attack can be expressed as:

$$\mathcal{A}(x^\star) = \mathbb{I}\left[f(x^\star) > \gamma\right] \tag{1}$$

where $\mathbb{I}$ is the indicator function, $f(x^\star)$ is a scoring function of $x^*$, and $\gamma$ is an adjustable decision threshold.

**Threat Model**    In this paper, we consider a threat model where the attacker has access to a set of synthetic data $S$ generated by a model $\mathcal{M}$ learned on $D$. We make no assumptions on the architecture or parameterization of the model nor do we assume the attacker has access to an API of the model in which to continuously query for an arbitrarily large $S$ (Meehan et al., 2020; Bhattacharjee et al., 2023). This corresponds to the practical scenario in which a synthetic dataset is released publicly for use. We also assume the attacker has access to a reference dataset $R$ that was not used in the training of the model, but is an independent sample from the same population as the training dataset, $T, R \overset{\text{iid}}{\sim} \mathbb{P}$. We assume this in practice because this represents a plausible scenario for the owner of $S$ as an attacker may be able to find comparable data in the real world such as open source datasets, paid collection, prior knowledge, etc. van Breugel et al. (2023) for example showed that reference datasets often improve the effectiveness of MIAs for generative models and many MIAs for supervised learning models incorporate reference sets as well in "shadow-box" attacks (Carlini et al., 2021; Ye et al., 2022; Zarifzadeh et al., 2024).

**Attack Strategy**    The adversary must develop a strategy in which to construct $f(x^*)$. We specifically propose that the adversary utilize the *degree of local overfitting* within $S$ as the primary signal to determine whether a specific data point $x^\star$ belongs to the training set.

Overfitting is a common and difficult-to-eliminate failure mode in generative models, particularly in the context of tabular synthetic data generation. In the setting of Membership Inference Attacks, this failure mode becomes a significant source of privacy leakage. van Breugel et al. (2023) for example identified that TVAE (Xu et al., 2019) overfit to minority class examples in a medical training dataset, leaking their privacy. Similarly, Ward et al. (2024) found that TabDDPM (Kotelnikov et al., 2022), when tasked with generating synthetic data for the well-known Adult dataset, heavily replicated data points from certain demographic groups within the training data. The key insight drawn from this phenomenon is that areas of the synthetic data distribution with higher density are likely to reflect signals from the original training data. Leveraging this failure, it becomes possible to infer whether specific data points were part of the training set, thus providing a basis for designing privacy attacks. Our work builds on these findings by proposing a new method to measure the degree of local overfitting in generative models. We utilize this metric to design a Membership Inference Attack aimed at exposing the potential privacy risks inherent in synthetic data (See Section 3).

## 3 GENERATIVE LIKELIHOOD RATIO ATTACK

In this section, we propose Generative Likelihood Ratio Attack (Gen-LRA), a powerful Membership Inference Attack which exploits overfitting to expose privacy leakage in generative models. Broadly speaking, Gen-LRA builds a hypothesis test around assessing if $S$ is overfit to $x^*$. By framing the problem as a hypothesis test, we can define a likelihood ratio that measures the extent of overfitting that is then used as the scoring function in Equation 1.

To begin, we compare the likelihood of the synthetic dataset $S$ under two competing hypotheses. The *null hypothesis* $H_0$ assumes that the synthetic data follows the population distribution $\mathbb{P}$, meaning that the generative model correctly models $\mathbb{P}$. Under this assumption, the likelihood of the synthetic dataset is given by:

$$H_0 : p(S|H_0) = \prod_{s \in S} p_\mathbb{P}(s) \tag{2}$$

In contrast, the *alternative hypothesis* $H_1$ assumes that the generative model overfits near $x^\star$, resulting in a modified probability distribution $p_{\mathbb{P} \cup \{x^\star\}}(s)$, which places additional weight on the vicinity

of $x^\star$. Thus, the likelihood of the synthetic dataset under $H_1$ is:

$$H_1 : p(S|H_1) = \prod_{s \in S} p_{\mathbb{P} \cup \{x^\star\}}(s) \tag{3}$$

This formulation suggests that the synthetic data distribution is overly influenced by $x^\star$, leading to a higher density of samples near this point. To compare the two hypotheses, we define the *likelihood ratio* as:

$$\lambda_{\mathbb{P}}(S, x^\star) = \frac{\prod_{s \in S} p_{\mathbb{P} \cup \{x^\star\}}(s)}{\prod_{s \in S} p_{\mathbb{P}}(s)} \tag{4}$$

However, equation 4 uses $\mathbb{P}$, meaning that the likelihood ratio $\lambda_{\mathbb{P}}(S, x^\star)$ operates at a *population level*. While theoretically well-defined, this ratio is computationally infeasible without full access to the population distribution. Instead, we use the reference dataset $R$ as an approximation of $\mathbb{P}$ as by definition from the threat model, $R \overset{\text{iid}}{\sim} \mathbb{P}$. We redefine the *sample-level* likelihood ratio as:

$$\lambda_R(S, x^\star) = \frac{\prod_{s \in S} p_{R \cup \{x^\star\}}(s)}{\prod_{s \in S} p_R(s)} \tag{5}$$

Here, $p_{R \cup \{x^\star\}}(s)$ represents the probability density of a synthetic element $s$ under the reference dataset augmented with $x^\star$. In contrast, $p_R(s)$ reflects the probability density under the reference dataset $R$ without the influence of $x^\star$. The intuition of this attack is that in the absence of overfitting (null hypothesis), the likelihood of the synthetic data should not significantly change with the inclusion of $x^*$ as an ideal generative model would produce synthetic data that follows the same population distribution as the training data. On the other hand, if overfitting is present (alternative hypothesis), the synthetic data will be concentrated near distinct points in the training set, leading to a distinct density increase around those points (See Figure 1).

### 3.1 GEN-LRA WITH KERNEL DENSITY ESTIMATORS

While $\lambda_R(S, x^\star)$ brings us closer to a practical computation compared to $\lambda_{\mathbb{P}}(S, x^\star)$, it remains computationally infeasible from observed data alone. Thus in order to implement Gen-LRA, we need to estimate the densities of $p_{R \cup \{x^\star\}}$ and $p_R$. While most density estimation techniques such as tractable probabilistic models (De Cao et al., 2019; Kobyzev et al., 2021; Liu & Van den Broeck, 2021) and Bayesian methods Grazian & Fan (2020); Hjort (1996) are compatible with Gen-LRA, we focus on studying Gen-LRA with non-parametric Gaussian Kernel Density Estimators (KDEs) (Weglarczyk, Stanisław, 2018) as they are widely known, computationally cheaper and have an explicit form. In our case, the KDE estimate for the density $\hat{p}_{R,K,h}(s)$ at a point $s$ is given by:

$$\hat{p}_{R,K,h}(s) = \frac{1}{nh} \sum_{i=1}^{n} K\left(\frac{s - r_i}{h}\right) \tag{6}$$

Here, $n$ is the number of samples in the reference dataset $R$, and $h$ is the bandwidth parameter that controls the smoothness of the estimate. The terms $r_i$ represent individual samples from the reference dataset $R$, and $K\left(\frac{s-r_i}{h}\right)$ is the kernel function applied to the scaled difference between the sample $s$ and the reference sample $r_i$. When incorporating the test point $x^\star$, the KDE for the augmented dataset $R \cup \{x^\star\}$ is given by:

$$\hat{p}_{R \cup \{x^\star\},K,h}(s) = \frac{1}{(n+1)h} \left[ \sum_{i=1}^{n} K\left(\frac{s - r_i}{h}\right) + K\left(\frac{s - x^\star}{h}\right) \right] \tag{7}$$

Thus, the likelihood ratio $\lambda_{R,K}(S, x^\star)$ can now be expressed as:

$$\lambda_{R,K}(S, x^\star) = \frac{\prod_{s \in S} \hat{p}_{R \cup \{x^\star\},K,h}(s)}{\prod_{s \in S} \hat{p}_{R,K,h}(s)} \tag{8}$$

Substituting the explicit KDE forms, we get:

$$\lambda_{R,K}(S, x^\star) = \frac{\prod_{s \in S} \left( \frac{1}{(n+1)h} \left[ \sum_{i=1}^{n} K\left(\frac{s-r_i}{h}\right) + K\left(\frac{s-x^\star}{h}\right) \right] \right)}{\prod_{s \in S} \left( \frac{1}{nh} \sum_{i=1}^{n} K\left(\frac{s-r_i}{h}\right) \right)} \tag{9}$$

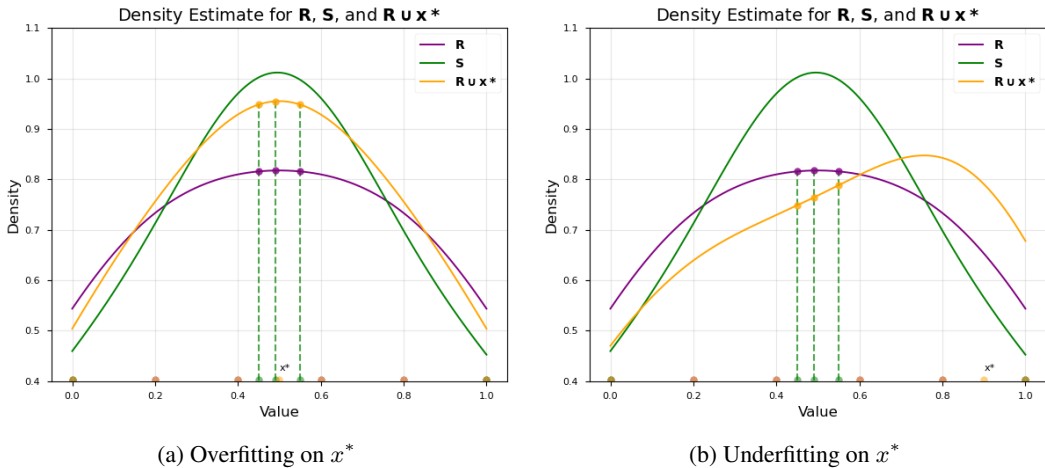

(a) Overfitting on $x^*$            (b) Underfitting on $x^*$

Figure 1: A geometric intuition for Gen-LRA with a 1-dimensional toy example. In Figures 1a and 1b, we visualize the KDE plots of $R, R \cup x^*, S$ as well as the estimated densities of the synthetic observations over $R$ and $R \cup x^*$. In Figure 1a we consider $x^* = 0.5$. Here, the likelihood of the synthetic observations (product of orange intersections) are higher under the density estimate of $R \cup x^*$ than $R$ (product of purple intersections) and therefore we conclude that $x^* \in T$. In Figure 1b where $x^* = 0.9$, the opposite is true and we therefore conclude $x^* \notin T$.

The likelihood ratio $\lambda_{R,K}(S, x^\star)$, as computed from the KDE-based estimates, serves as our **scoring function** for membership prediction:

$$f(x^\star) \equiv \lambda_{R,K}(S, x^\star) \tag{10}$$

By computing this score, we measure the degree of local overfitting around the test point $x^\star$. A higher score indicates that the synthetic dataset is likely overfitting near $x^\star$, suggesting that this point was part of the training data used to generate the synthetic samples. Thus, by thresholding $\lambda_{R,K}(S, x^\star)$, we can perform *membership inference*—predicting whether $x^\star$ belongs to the original training dataset. This method allows us to use the synthetic dataset to infer sensitive information about the underlying training data with no assumptions of the qualities of the generative model that generated it.

### 3.2 GEN-LRA IMPLEMENTATION

We refer to Algorithm 1 for a pseudo-code description of the Gen-LRA attack.

**Localization** A common theme in designing MIAs is to adopt techniques that maximize the signal of $x^*$'s membership in the attack. Realistically, there is likely to be very little signal in comparing the likelihoods of $S$ over estimated probability density functions with a difference of a single observation. Indeed, equation 5 is an attack over the global difference in $p_{R \cup x^*}$ and $p_R$. Instead, if we have some idea for what regions of the two estimated PDFs are likely to most differ, we can focus the attack on areas that should contain the most signal. Here, we *localize* Gen-LRA by only considering the $k$-nearest elements in $S$ to $x^*$ in our calculation of equation 5. In practice, the choice of $k$ can have minor impacts on the effectiveness of the attack, but we find we get excellent results with small values (See Appendix A.3).

**Choice of Decision Threshold** While the previous sections detail the derivation of a scoring function, equation 1 still requires a decision threshold $\gamma$. Intuitively for Gen-LRA, $\gamma$ can be any chosen threshold but $\lambda_{R,K}(S, x^\star > 1$ implies some degree of overfitting. As many MIAs do not have a natural thresholding heuristic, a technique often employed is simply taking the median score over many test observations. In practice though, as MIAs are privacy auditing tools the decision threshold is less important as a practitioner should evaluate the attack at all possible thresholding values to understand the maximal privacy risk of the attack. In evaluating MIAs, we therefore focus on

metrics like AUC-ROC and True Positive Rate at False Positive Rate as these are independent of a fixed $\gamma$ (See Section 5 for more details).

---

**Algorithm 1** Gen-LRA

---

**Require:**
1: $\mathbf{X}_{\text{test}} \in \mathbb{R}^{n_{\text{test}} \times d}$: Test dataset
2: $\mathbf{S} \in \mathbb{R}^{n_S \times d}$: Generated dataset
3: $\mathbf{R} \in \mathbb{R}^{n_{\text{ref}} \times d}$: Reference dataset
4: $k \in \mathbb{N}$: Number of closest points to compare
**Ensure:**
5: $\mathbf{S}_{\text{scores}} \in \mathbb{R}^{n_{\text{test}}}$: Attack scores for test samples
6: **function** GENLRATTACK($\mathbf{X}_{\text{test}}, \mathbf{S}, \mathbf{R}, k$)
7: $\quad$ $\mathbf{S}_{\text{scores}} \leftarrow \emptyset$ $\hfill \triangleright$ Initialize score array
8: $\quad$ $\text{DE}_{\mathbf{R}} \leftarrow \text{FitDensityEstimator}(\mathbf{R})$ $\hfill \triangleright$ Fit density estimator on $\mathbf{R}$
9: $\quad$ **for** $\mathbf{x} \in \mathbf{X}_{\text{test}}$ **do**
10: $\quad\quad$ $\mathbf{R}' \leftarrow \mathbf{R} \cup \{\mathbf{x}\}$ $\hfill \triangleright$ Insert $\mathbf{x}$ into reference set
11: $\quad\quad$ $\text{DE}_{\mathbf{R}'} \leftarrow \text{FitDensityEstimator}(\mathbf{R}')$ $\hfill \triangleright$ Fit density estimator on $\mathbf{R}'$
12: $\quad\quad$ $\mathbf{S}_{\text{close}} \leftarrow \text{FindKNearestNeighbors}(\mathbf{S}, \mathbf{x}, k)$ $\hfill \triangleright$ Find $k$ closest points in $\mathbf{S}$
13: $\quad\quad$ $\mathbf{L}_{\mathbf{R}'} \leftarrow \text{DE}_{\mathbf{R}'}(\mathbf{S}_{\text{close}})$ $\hfill \triangleright$ Compute likelihoods using $\text{DE}_{\mathbf{R}'}$
14: $\quad\quad$ $\mathbf{L}_{\mathbf{R}} \leftarrow \text{DE}_{\mathbf{R}}(\mathbf{S}_{\text{close}})$ $\hfill \triangleright$ Compute likelihoods using $\text{DE}_{\mathbf{R}}$
15: $\quad\quad$ $s \leftarrow \sum_{\mathbf{s} \in \mathbf{S}_{\text{close}}} \log(\mathbf{L}_{\mathbf{R}'}[\mathbf{s}]) - \sum_{\mathbf{s} \in \mathbf{S}_{\text{close}}} \log(\mathbf{L}_{\mathbf{R}}[\mathbf{s}])$ $\hfill \triangleright$ Compute log-likelihood difference
16: $\quad\quad$ $\mathbf{S}_{\text{scores}} \leftarrow \mathbf{S}_{\text{scores}} \cup \{s\}$
17: $\quad$ **end for**
18: $\quad$ **return** $\mathbf{S}_{\text{scores}}$
19: **end function**

---

## 4 RELATED WORKS

### 4.1 ASSESSING OVERFITTING IN TABULAR GENERATIVE MODELS

Several measures have been developed to assess the fitness of tabular synthetic data, particularly from a privacy perspective. These metrics generally aim to measure the similarity between the training and synthetic datasets, with the ideal outcome being that the synthetic data is neither too similar to the training data nor too different. A widely used metric for this purpose is Distance to Closest Record[1] (Park et al., 2018; Lu et al., 2019; Yale et al., 2019; Zhao et al., 2021; Guillaudeux et al., 2022; Liu et al., 2023), which compares the distance from each training point to its nearest neighbor in the synthetic dataset to which a mean is computed. Another commonly used metric is the Identical Matching Score (IMS) (Lu et al., 2019; AI, 2020; 2021), which measures the proportion of identical records between the training and synthetic datasets. While these measures can be useful for describing overfitness from a sample quality and model generalization perspective, they do not characterize privacy risk because there is no assumed threat model and they are not evaluated over non-member examples.

### 4.2 MIAS FOR MACHINE LEARNING MODELS

Membership Inference Attacks on the other hand, explicitly characterize the empirical privacy risk of a machine learning model (Song & Mittal, 2020; Yeom et al., 2018). Originally, MIAs were developed for attacking supervised learning classifiers (Shokri et al., 2017). In this context, the general idea for these attacks is to query a model with different observations to learn patterns in its class probability outputs. Membership can then be inferred by comparing the outputs of the model to outputs from reference models in some manner (Sablayrolles et al., 2019a; Long et al., 2020; Carlini et al., 2021; Watson et al., 2022; Ye et al., 2022; Zarifzadeh et al., 2024). While fundamental to the literature, these methods are largely incompatible with attacking synthetic data generators as

---

[1]DCR in the similarity metric case compares a training point to a synthetic point. However, Chen et al. (2020) proposes an MIA where the scoring function is a distance computation for a test point and a synthetic point. In all other sections of the paper we use DCR to refer to the MIA.

they rely on unlimited query access to the model and also formulate their attacks around returned probability predictions.

To adapt to the structural differences in the problem domain, MIAs in the generative model setting have adopted two broad styles of developing a scoring function: distance-based and density-based attacks. Distance-based attacks rely on using some measure of distance between the test observation and the synthetic and/or reference sets (Hayes et al., 2017; Chen et al., 2020; Ward et al., 2024). Similarly to Gen-LRA, density-based attacks attack inconsistencies in the probability densities of the synthetic and reference sets (Hilprecht et al., 2019; van Breugel et al., 2023). While these works usually cover MIAs under a wide range of threat models, we only consider the attacks that use a black-box (only synthetic data access) or shadow-box (only synthetic and reference data access) threat model. This is in contrast to white-box attacks in which an adversary have both synthetic and reference data as well as internal access to the model (Matsumoto et al., 2023; Pang et al., 2023; Wu et al., 2023). White-box attacks generally are specific to the architecture of a model (Sablayrolles et al., 2019b) and are not generalizable to the broader synthetic data release paradigm in which data owners usually do not release their model weights.

## 5 EXPERIMENTS

### 5.1 BENCHMARKING

We test the effectiveness of Gen-LRA across a benchmark of 15 tabular datasets (Full details on MIAs, architectures, and datasets are in Appendix A.4, A.5, A.6, respectively). From each dataset, we scale continuous and one-hot-encode discrete variables before randomly sampling without replacement 3 equal sized training $T$, reference $R$, and holdout testing $H$ sets. The training set is used to train various popular private and non-private architectures to which an equally sized synthetic set is generated. Using the synthetic and reference sets, MIAs are then evaluated by their AUC-ROC and Accuracy on distinguishing between the training and holdout testing sets $X^* = T \cup H$. We repeat this 10 times for each dataset for each $T, R, H$, and $S$ with sample size of $N = (250, 1000, 4000)$.

For DOMIAS and Gen-LRA which rely on density estimation, we implement these methods using a Kernel Density Estimator (KDE). As KDEs struggle to converge for high dimensional, heterogeneous data, in line with Wen & Hang (2022), we reduce the dimensionality for qualifying datasets by fitting a Principle Component Analysis (PCA) transformation to $S$ and transforming $R$ and $X^*$ accordingly.

The full results for each MIA's mean and standard deviation AUC-ROC across all runs and $N$-sizes for each architecture are reported in table 1. We report a similar table for accuracy in Appendix A.1.2 although the results are largely equivalent. For Gen-LRA, we found that the choice of $k$ can have a small impact on the performance of the attack (See Appendix A.3), we therefore use the results of the best $k$ choice for each run as the goal for an MIA is to characterize the maximal empirical privacy risk.

Table 1: Average AUC-ROC for each Membership Inference Attack across model architectures and datasets.

| Model | Gen-LRA (Ours) | DCR-Diff | DPI | DOMIAS | DCR | MC | Logan 2017 |
|---|---|---|---|---|---|---|---|
| AdsGAN | **0.529 (0.02)** | 0.517 (0.02) | 0.521 (0.02) | 0.517 (0.02) | 0.516 (0.02) | 0.515 (0.02) | 0.503 (0.02) |
| ARF | **0.548 (0.03)** | 0.540 (0.02) | 0.538 (0.02) | 0.534 (0.02) | 0.533 (0.02) | 0.527 (0.02) | 0.504 (0.02) |
| Bayesian Network | 0.654 (0.07) | 0.656 (0.06) | 0.557 (0.02) | 0.632 (0.06) | **0.680 (0.07)** | 0.625 (0.05) | 0.505 (0.02) |
| CTGAN | **0.527 (0.02)** | 0.515 (0.02) | 0.519 (0.02) | 0.515 (0.02) | 0.513 (0.02) | 0.511 (0.02) | 0.504 (0.02) |
| Tab-DDPM | **0.603 (0.08)** | 0.587 (0.06) | 0.552 (0.03) | 0.587 (0.06) | 0.585 (0.07) | 0.564 (0.05) | 0.505 (0.02) |
| Normalizing Flows | **0.517 (0.02)** | 0.504 (0.02) | 0.506 (0.02) | 0.505 (0.02) | 0.505 (0.02) | 0.504 (0.02) | 0.502 (0.02) |
| PATEGAN | **0.514 (0.02)** | 0.497 (0.02) | 0.500 (0.02) | 0.498 (0.02) | 0.500 (0.02) | 0.501 (0.02) | 0.502 (0.02) |
| TVAE | **0.541 (0.02)** | 0.529 (0.03) | 0.523 (0.02) | 0.524 (0.03) | 0.529 (0.03) | 0.522 (0.02) | 0.504 (0.02) |
| **Rank** | **1.3** | 3.5 | 3.6 | 3.8 | 4.0 | 5.4 | 6.4 |

Overall, Gen-LRA outperforms other MIAs across nearly all architectures with an average rank of 1.3. As Gen-LRA relies on estimating the likelihood of high dimensional, heterogeneous data, it is surprising that it excels with using PCA coupled with KDE, which is a baseline that is usually beaten by more modern density estimation methods (Wen & Hang, 2022; De Cao et al., 2019). Although using these newer methods would likely improve the attack, we benchmark with PCA + KDE as it

is computationally cheaper than these methods and it showcases that the gain in the attack comes from equation 5, minimally implemented.

## 5.2 THE LOW FALSE POSITIVE SETTING

While AUC-ROC provides an easily comparable global measure of an attack's effectiveness, from a privacy perspective it does not indicate how well an attack performs when the False Positive Rate (FPR) is low. As Carlini et al. (2021) and Zarifzadeh et al. (2024) argue, researchers should analyze how well an attack performs with a low FPR because in practical settings there is a greater privacy risk to individual training observations that can be correctly classified with few false positives versus observations that are included with many false positives. Similarly, as the goal of MIAs is to identify positive membership, identifying if $x^*$ *is not* a member is less important.

We therefore report the mean and standard deviation TPR@FPRs (True Positive Rate at False Positive Rate) for a range of fixed FPR values for each MIA across datasets, architectures, and $N$-sizes in table 2. Achieving a high TPR at a very low FPR is challenging in this scenario as MIAs are inherently an unsupervised classification task. However, Gen-LRA nearly doubles the performance of the next best method at FPR = 0.001 and consistently sees significant gains over the next best method at higher thresholds. This highlights that Gen-LRA is better able to detect egregious overfitting to certain training observations, relative to other competing attacks.

Table 2: True Positive Rates for MIAs at different fixed False Positive Rate levels.

| MIA | 0.001 | 0.01 | 0.1 |
|---|---|---|---|
| Logan 2017 | 0.003 (0.01) | 0.012 (0.01) | 0.102 (0.02) |
| DPI | 0.002 (0.00) | 0.014 (0.01) | 0.118 (0.03) |
| MC | 0.003 (0.00) | 0.014 (0.01) | 0.120 (0.04) |
| DOMIAS | 0.002 (0.00) | 0.016 (0.01) | 0.134 (0.06) |
| DCR-Diff | 0.005 (0.01) | 0.019 (0.02) | 0.138 (0.07) |
| DCR | 0.016 (0.05) | 0.036 (0.08) | 0.153 (0.11) |
| Gen-LRA (ours) | **0.031 (0.01)** | **0.056 (0.03)** | **0.193 (0.08)** |

## 5.3 TARGETING OVERFITTING IN OUTLIERS

An additional question we are interested in investigating is if Gen-LRA displays patterns of behavior that are different from other MIAs. As a case study, we replicate an experiment from Ward et al. (2024) where the authors train Tab-DDPM on the Adult dataset, and evaluate Membership Inference Attack scores over a 2-D projection of the training set. Here, we perform this same procedure, plotting a UMAP projection (McInnes et al., 2018) of the training data and coloring the observations with the 99.5th percentile highest Gen-LRA and DCR scores (See figure 2).

We find that Gen-LRA's highest scores are concentrated in an outlier cluster in the (0,12) region whereas DCR's are spread through the plot. We examined the observations in this cluster and found that nearly every data point had the *same* demographics: white, male, American, married, high income, and with high capital gains. This provides evidence that Gen-LRA is specifically attacking overfitting to outlier regions of the training distribution.

## 6 DISCUSSION

### 6.1 GEN-LRA PERFORMANCE

Gen-LRA is a density-based attack that, using a simple estimation strategy, broadly outperforms competing methods. Constructing the attack as a likelihood ratio over local regions of the synthetic probability distribution allows greater attack performance as Gen-LRA is customizable in its choice of $k$ to different datasets and architectures. Indeed as table 1 shows, models like Tab-DDPM and Bayesian Networks experience more privacy leakage than others and a tunable attack can realize large performance gains. While Gen-LRA excels in a global attack evaluation setting with an average rank of 1.3 across models, Gen-LRA also sees best-in-class performance in the more difficult

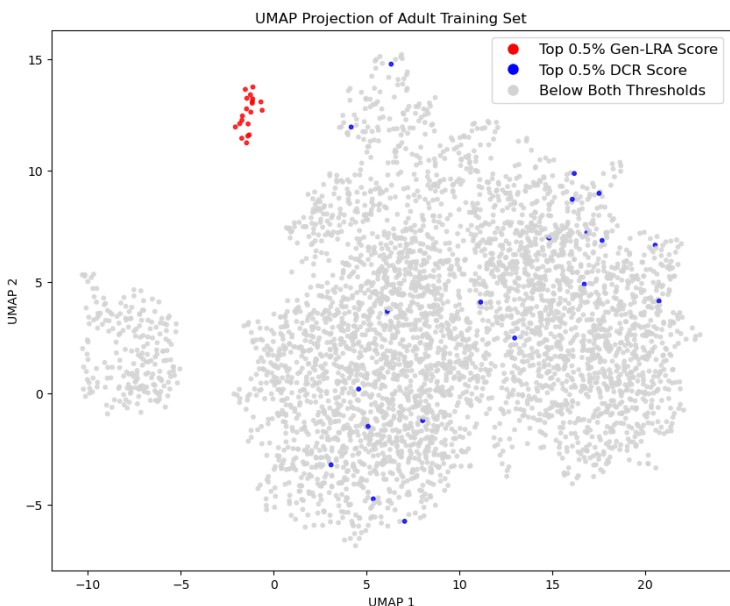

Figure 2: A UMAP projection of the Adult training dataset then used to train Tab-DDPM. In red and blue are the observations with 99.5th percentile Gen-LRA and DCR scores respectively. Gen-LRA targets specific outlier regions of the distribution whereas DCR is dispersed. Concerningly, the cluster at (0,12) are nearly all observations with white, male, American, married, high income, and high capital gains demographics. This suggests that specific subgroups of training data can experience more privacy leakage than others.

low FPR setting. While TPR@FPR performance for generative MIAs is lower than in the supervised setting, table 2 indicates Gen-LRA is a step in the right direction as most other attacks outright fail at the 0.001 and 0.01 levels. Lastly, figure 2 shows Gen-LRA attacks outlier regions of a training distribution. This is surprising as Gen-LRA can indicate where and how a generative model may be overfitting to its training data and it highlights that the privacy leakage of individuals appears non-random in that similar training observations can be more egregiously overfit to than others.

### 6.2 ON DISTANCE VERSUS DENSITY-BASED ATTACKS

One finding is that distance based attacks like DCR can outperform density based attacks like Gen-LRA in some architectures and datasets. For example, DCR slightly outperforms Gen-LRA with Bayesian Networks and is the next best method in the low FPR domain. We hypothesize that this is because DCR and Gen-LRA attack fundamentally different types of overfitting. Consider two toy data simulations (full details in A.2.1): in one we let $T$ and $R$ be random samples from a 2-dimensional standard multivariate Gaussian: $T, R \overset{iid}{\sim} \mathcal{N}_2(\mathbf{0}, \mathbf{I})$ and a model $\mathcal{M}$ exactly copies training examples for its output; $S = T$. In the other, we similarly let $R \overset{iid}{\sim} \mathcal{N}_2(\mathbf{0}, \mathbf{I})$ but, the sampling distribution of $T$ is made to slightly differ than $R$ (perhaps due to sampling variation or bias) and $S$ well-models $T$ such that $D, S \overset{iid}{\sim} \mathcal{N}_2(\mathbf{0}, \begin{pmatrix} 2 & 0 \\ 0 & 1 \end{pmatrix})$. The average AUC-ROC of DCR and Gen-LRA are compared in table 3.

For the data copying simulation, distance based attacks like DCR always outperform density attacks because all measures of distance between $T$ and $S$ are 0. A DCR MIA will thus always have an Accuracy and AUC-ROC of 1 and Gen-LRA struggles to detect any privacy leakage. On the other hand, DCR is worse than random at detecting privacy leakage from a generator overfitting to a training dataset, relative to the reference population distribution whereas Gen-LRA identifies this difference. In practice, there will usually be natural variation between the empirical distributions of $P_T$ and $P_R$, the danger that Gen-LRA highlights is that $G$ can leak the privacy of training data by generating $S$ that is closer to $p_T$ than to the true population distribution. Indeed, this is further

evidenced by figure 2 that demonstrates Gen-LRA attacks specific outlier regions of distributions whereas DCR does not.

Table 3: AUC-ROC for MIAs across the data copying and overfitting toy simulations.

| Simulation Example | DCR | Gen-LRA |
|---|---|---|
| Data Copying | **1.00 (.00)** | 0.53 (.02) |
| Overfitting | 0.46 (.02) | **0.59 (.02)** |

## 7 CONCLUSION

Membership Inference Attacks are a useful tool for evaluating generative models for synthetic data release. They can characterize the privacy risk towards training observations, provide information on how a model may be overfit, and add subtle context to patterns of behavior in generative models. In this paper, we propose Gen-LRA, which constructs a likelihood ratio of the synthetic data using simple Kernel Density Estimators. We show that it excels at attacking a diverse set of generative models across a wide-range of datasets and that this success comes from Gen-LRA's unique ability to target a generative model's tendency to overfit to training outliers- a trait that is not well-shared with other common MIAs. We note that there are several drawbacks with Gen-LRA in that it requires dimension reduction techniques to be compatible with high dimensional heterogeneous data and that it fails at detecting flagrant data-copying. However, we point out that Gen-LRA is compatible with high dimensional density estimation strategies and that empirically, Gen-LRA usually outperforms other attacks despite these disadvantages.

We believe that there are many directions for future work. Exploring emerging density estimation methodologies would likely yield better empirical performance, especially on high dimensional datasets. On a different front, research into developing adversarial techniques to better understand model overfitting in general could also lead to important interpretability techniques. Lastly, we believe that it is important to investigate the observed phenomenon that Gen-LRA can specifically target distinct sub-groups of a training dataset as this implies that even if an attack is largely unsuccessful in the aggregate, high-risk observations may still be leaked.

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

# A  APPENDIX

## A.1  ADDITIONAL FIGURES

### A.1.1  SAMPLE SIZE AND MIA EFFECTIVENESS

It is known that Membership Inference Attacks benefit from low sample sizes of $T$, $R$, and $S$. We explore the effect of the size of these samples across all models and datasets in figure 3. Here, we see that performance drops off between $N$=250 and $N$=1000; however it is relatively the same across all MIAs between $N$=1000 and $N$=4000. Across all N-sizes, Gen-LRA has a greater average AUC-ROC then all other MIAs. This further demonstrates that Gen-LRA is an excellent choice for a privacy auditing adversarial attack.

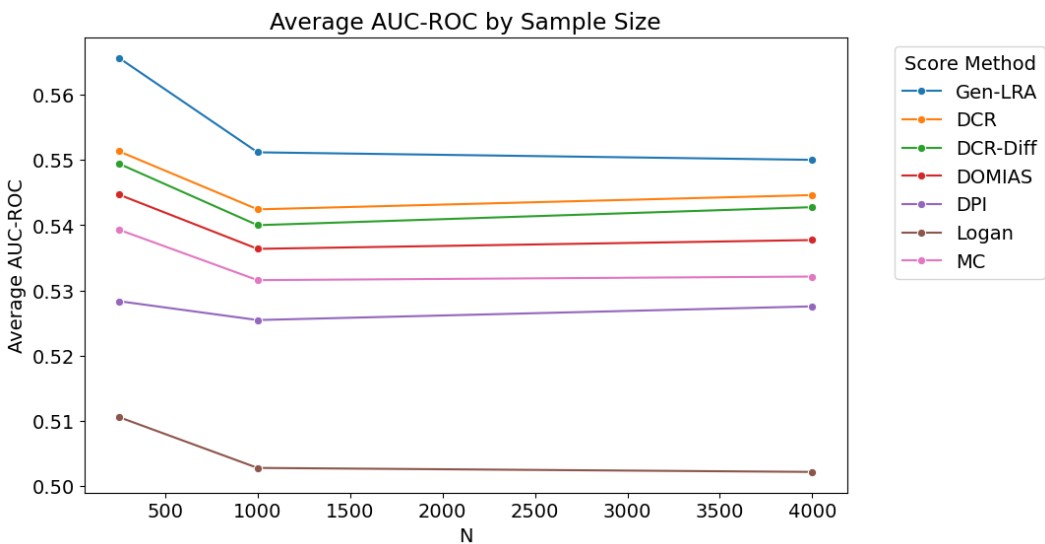

Figure 3: Average MIA AUC-ROC across different sample sizes. There is little decrease in performance after $N$=1000 and Gen-LRA has the highest global attack performance across N-sizes.

### A.1.2  AVERAGE ACCURACY TABLE

Table 4: Average AUC-ROC for each Membership Inference Attack across model architectures and datasets.

| Model | Gen-LRA (Ours) | MC | DCR | DCR-Diff | DPI | DOMIAS | LOGAN 2017 |
|---|---|---|---|---|---|---|---|
| AdsGAN | **0.524 (0.02)** | 0.513 (0.02) | 0.513 (0.02) | 0.513 (0.02) | 0.515 (0.02) | 0.513 (0.02) | 0.503 (0.02) |
| ARF | **0.539 (0.02)** | 0.524 (0.02) | 0.524 (0.02) | 0.529 (0.02) | 0.526 (0.02) | 0.524 (0.02) | 0.503 (0.02) |
| Bayesian Network | 0.619 (0.05) | **0.629 (0.05)** | 0.629 (0.05) | 0.621 (0.05) | 0.538 (0.02) | 0.599 (0.05) | 0.504 (0.02) |
| CTGAN | **0.523 (0.02)** | 0.509 (0.02) | 0.509 (0.02) | 0.511 (0.02) | 0.513 (0.02) | 0.511 (0.02) | 0.504 (0.02) |
| Tab-DDPM | **0.58 (0.04)** | 0.564 (0.05) | 0.564 (0.05) | 0.563 (0.05) | 0.537 (0.02) | 0.563 (0.04) | 0.504 (0.02) |
| Normalizing Flows | **0.517 (0.02)** | 0.504 (0.02) | 0.504 (0.02) | 0.504 (0.02) | 0.505 (0.02) | 0.504 (0.02) | 0.501 (0.02) |
| PATEGAN | **0.514 (0.02)** | 0.501 (0.02) | 0.501 (0.02) | 0.499 (0.02) | 0.499 (0.02) | 0.500 (0.02) | 0.501 (0.02) |
| TVAE | **0.533 (0.02)** | 0.520 (0.02) | 0.520 (0.02) | 0.522 (0.02) | 0.517 (0.02) | 0.518 (0.02) | 0.503 (0.02) |
| **Rank** | **1.3** | 3.2 | 3.4 | 3.6 | 3.6 | 3.9 | 5.5 |

### A.1.3  MODEL UTILITY AND GEN-LRA EFFECTIVENESS

We benchmark various statistical metrics used to describe the quality of tabular synthetic data across architectures and datasets. We plot the mean Wasserstein distance and Maximum Mean Discrepancy between the corresponding training and synthetic data against the mean AUC-ROC of Gen-LRA in figure 4. Here, it seems there is some relationship between measures of statistical distance and Gen-LRA's global effectiveness. As these metrics are often used in utility benchmarks for tabular synthetic data, it is important to note that for practitioners, statistical fidelity in synthetic data can

come at a privacy cost. It also illustrates that measures of utility should include some kind of holdout testing method to consider overfitting.

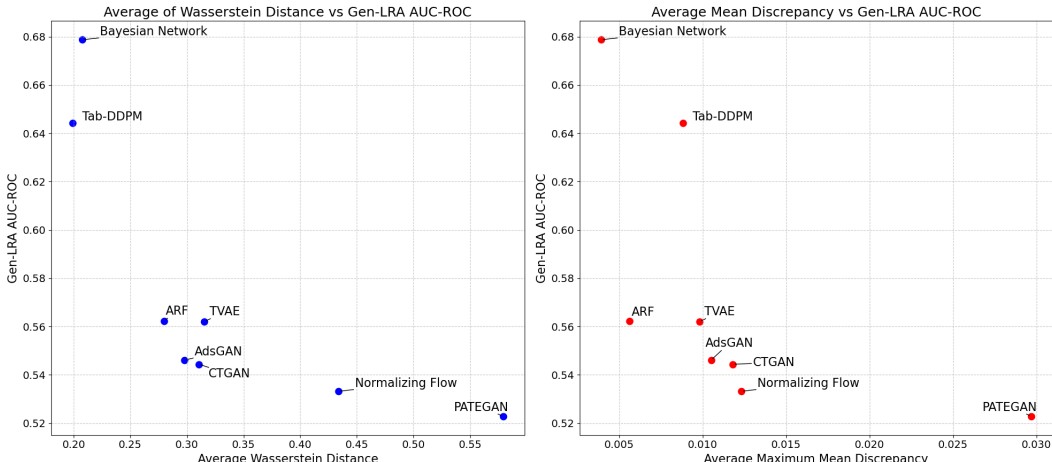

Figure 4: Average Wasserstein Distance and Average Maximum Mean Discrepancy plotted against Gen-LRA AUC-ROC for benchmarked models. Bayesian Network and Tab-DDPM outperform other models in these performance metrics but have higher privacy risk.

## A.2   EXPERIMENT DETAILS

### A.2.1   SECTION 6.2

We conducted two experiments to evaluate the performance of DCR and Gen-LRA on different types of model failure, with the full results shown in table 3. The experiments were carried out as follows:

**Data Copying Simulation**   In this setup, we let $T$ and $R$ be random samples from a 2-dimensional standard multivariate Gaussian distribution; i.e., $T, R \overset{\text{iid}}{\sim} \mathcal{N}_2(\mathbf{0}, \mathbf{I})$. Here, we assume a model $\mathcal{M}$ that exactly reproduces the training examples in its output, meaning $S = T$.

**Overfitting Simulation**   In this simulation, we again let $R \overset{\text{iid}}{\sim} \mathcal{N}_2(\mathbf{0}, \mathbf{I})$, but the sampling distribution of $T$ is modified to slightly differ from $R$, potentially due to sampling variation or bias. In this case, the output $S$ models $T$ well, where $D, S \overset{\text{iid}}{\sim} \mathcal{N}_2(\mathbf{0}, \begin{pmatrix} 2 & 0 \\ 0 & 1 \end{pmatrix})$.

For both simulations, we set the sample size $n = 500$ for $T$, $R$, and $S$, and the AUC-ROC of DCR and Gen-LRA was compared over 10,000 iterations.

## A.3   ABLATION: DIFFERENT $k$ SIZES

Gen-LRA targets local fitting by selecting a subset of $S$ to evaluate likelihoods with. This is implemented using the $k$-nearest neighbors in $S$ to $x^*$. In practice, this means that $k$ must be selected as a hyperparameter for the attack. In order to understand how $k$ impacts the quality of the attack, we replicate section 5 benchmarking with various $k$ values. We report the average AUC-ROC and standard deviations in table 5. Overall, we find that empirically usually smaller values of $k$ are better although it depends on the model. As stated in section 3, a global attack over all $S$ is unlikely to yield much membership signal. This is confirmed with $k = N$, where the AUC-ROC is always 0.5 and highlights that overfitting is a local phenomenon and that generative model adversarial attacks should focus on attacking locality to be successful.

Table 5: Average AUC-ROC at different $k$ values for Gen-LRA.

| Model | k=1 | k=3 | k=5 | k=10 | k=15 | k=20 | k=N |
|---|---|---|---|---|---|---|---|
| AdsGAN | 0.514 (0.02) | 0.518 (0.02) | 0.519 (0.02) | 0.520 (0.02) | 0.521 (0.02) | 0.521 (0.02) | 0.500 (0.00) |
| ARF | 0.532 (0.02) | 0.538 (0.02) | 0.540 (0.02) | 0.540 (0.03) | 0.540 (0.03) | 0.539 (0.03) | 0.500 (0.00) |
| Bayesian Network | 0.650 (0.07) | 0.645 (0.07) | 0.640 (0.07) | 0.634 (0.07) | 0.631 (0.07) | 0.629 (0.07) | 0.500 (0.00) |
| CTGAN | 0.514 (0.02) | 0.516 (0.02) | 0.517 (0.02) | 0.517 (0.02) | 0.518 (0.02) | 0.518 (0.02) | 0.500 (0.00) |
| Tab-DDPM | 0.595 (0.07) | 0.595 (0.07) | 0.594 (0.07) | 0.592 (0.06) | 0.591 (0.06) | 0.589 (0.06) | 0.500 (0.00 |
| Normalizing Flow | 0.503 (0.02) | 0.503 (0.02) | 0.505 (0.02) | 0.506 (0.02) | 0.506 (0.02) | 0.506 (0.02) | 0.500 (0.00) |
| TVAE | 0.527 (0.03) | 0.531 (0.03) | 0.531 (0.03) | 0.531 (0.03) | 0.530 (0.03) | 0.529 (0.03) | 0.500 (0.00) |

## A.4 MIAs FOR GENERATIVE MODELS DESCRIPTIONS

The Membership Inference Attacks referenced in this paper is are described as follows:

- **LOGAN** Hayes et al. (2017): LOGAN consists of black box and shadow box attack. The black-box version involves training a Generative Adversarial Network (GAN) on the synthetic dataset and using the discriminator to score test data. A calibrated version improves upon this by training a binary classifier to distinguish between the synthetic and reference dataset. In this paper, we only benchmark the calibrated version.

- **Distance to Closest Record (DCR) / DCR Difference** Chen et al. (2020): DCR is a black-box attack that scores test data based on a sigmoid score of the distance to the nearest neighbor in the synthetic dataset. DCR Difference enhances this approach by incorporating a reference set, subtracting the distance to the closest record in the reference set from the synthetic set distance.

- **MC** Hilprecht et al. (2019): MC is based on counting the number of observations in the synthetic dataset that fall into the neighborhood of a test point (Monte Carlo Integration). However, this method does not consider a reference dataset, and the choice of distance metric for defining a neighborhood is a non-trivial hyperparameter to tune.

- **DOMIAS** van Breugel et al. (2023): DOMIAS is a calibrated attack which scores test data by performing density estimation on both the synthetic and reference datasets. It then calculates the density ratio of the test data between the learned synthetic and reference probability densities.

- **DPI** Ward et al. (2024): DPI computes the ratio of $k$-Nearest Neighbors of $x^*$ in the synthetic and reference datasets. It then builds a scoring function by computing the ratio of the sum of data points from each class of neighbors from the respective sets.

## A.5 GENERATIVE MODEL ARCHITECTURE DESCRIPTIONS

In all experiments, we use the implementations of these models from the Python package Synthcity Qian et al. (2023). For benchmarking purposes, we use the default hyperparameters for each model. A brief description of each model is as follows:

- **CTGAN** Xu et al. (2019): Conditional Tabular Generative Adversarial Network uses a GAN framework with conditional generator and discriminator to capture multi-modal distributions. It employs mode normalization to better learn mixed-type distributions.

- **TVAE** Xu et al. (2019): Tabular Variational Auto-Encoder is similar to CTGAN in its use of mode normalizing techniques, but instead of a GAN architecture, it employs a Variational Autoencoder.

- **Normalizing Flows (NFlows)** Durkan et al. (2019): Normalizing flows transform a simple base distribution (e.g., Gaussian) into a more complex one matching the data by applying a sequence of invertible, differentiable mappings.

- **Bayesian Network (BN)** Ankan & Panda (2015): Bayesian Networks use a Directed Acyclic Graph to represent the joint probability distribution over variables as a product of marginal and conditional distributions. It then samples the empirical distributions estimated from the training dataset.

- **Adversarial Random Forests (ARF)** Watson et al. (2023): ARFs extend the random forest model by adding an adversarial stage. Random forests generate synthetic samples which

are scored against the real data by a discriminator network. This score is used to re-train the forests iteratively.

- **Tab-DDPM** Kotelnikov et al. (2022): Tabular Denoising Diffusion Probabilistic Model adapts the DDPM framework for image synthesis. It iteratively refines random noise into synthetic data by learning the data distribution through gradients of a classifier on partially corrupted samples with Gaussian noise.

- **PATEGAN** Yoon et al. (2019): The PATEGAN model uses a neural encoder to map discrete tabular data into a continuous latent representation which is sampled from during generation by the GAN discriminator and generator pair.

- **Ads-GAN** Yoon et al. (2020b): Ads-GAN uses a GAN architecture for tabular synthesis but also adds an identifiability metric to increase its ability to not mimic training data.

## A.6 BENCHMARKING DATASETS REFERENCES

We provide the URL for the sources of each dataset considered in the paper. We use datasets common in the tabular generative modeling literature Suh et al. (2023)

1. **Abalone** (OpenML): `https://www.openml.org/search?type=data&sort=runs&id=183&status=active`

2. **Adult** Becker & Kohavi (1996)

3. **Bean** (UCI): `https://archive.ics.uci.edu/dataset/602/dry+bean+dataset`

4. **Churn-Modeling** (Kaggle): `https://www.kaggle.com/datasets/shrutimechlearn/churn-modelling`

5. **Faults** (UCI): `https://archive.ics.uci.edu/dataset/198/steel+plates+faults`

6. **HTRU** (UCI): `https://archive.ics.uci.edu/dataset/372/htru2`

7. **Indian Liver Patient** (Kaggle): `https://www.kaggle.com/datasets/uciml/indian-liver-patient-records?resource=download`

8. **Insurance** (Kaggle): `https://www.kaggle.com/datasets/mirichoi0218/insurance`

9. **Magic** (Kaggle): `https://www.kaggle.com/datasets/abhinand05/magic-gamma-telescope-dataset?resource=download`

10. **News** (UCI): `https://archive.ics.uci.edu/dataset/332/online+news+popularity`

11. **Nursery** (Kaggle): `https://www.kaggle.com/datasets/heitornunes/nursery`

12. **Obesity** (Kaggle): `https://www.kaggle.com/datasets/tathagatbanerjee/obesity-dataset-uci-ml`

13. **Shoppers** (Kaggle): `https://www.kaggle.com/datasets/henrysue/online-shoppers-intention`

14. **Titanic** (Kaggle): `https://www.kaggle.com/c/titanic/data`

15. **Wilt** (OpenML): `https://www.openml.org/search?type=data&sort=runs&id=40983&status=active`

