# OpenReview forum: "Gen-LRA: Towards a Principled Membership Inference Attack for Generative Models"
_ICLR.cc/2025/Conference — ICLR 2025 Conference Withdrawn Submission_

### Official Review · Reviewer_Phn8 · 2024-10-28

**Soundness:** 2
**Presentation:** 3
**Contribution:** 3
**Rating:** 5
**Confidence:** 4

**Summary:**

The paper proposes a novel membership inference attack on synthetic data generators called Gen-LRA, based on estimating a likelihood ratio between the synthetic data coming from a reference distribution vs. it coming from the reference distribution with a target point included. Gen-LRA is benchmarked againt several competing attacks on a variety of datasets, where Gen-LRA generally outperforms the competition.

**Strengths:**

The likelihood ratio that Gen-LRA estimates is novel to my knowledge, and seems to be closer to the likelihood ratio that would be theoretically optimal than what previous work has looked at. The paper is easy to understand, and the writing is generally polished.

Looking at TPR @ low FPR is good practice, and too often neglected in the MIA literature. The paper could even highlight these results further: most of the AUC-ROC scores for all methods are close to random guessing, but Gen-LRA is much more accurate than random guessing at FPR = 0.001.

**Weaknesses:**

Using the PCA+KDE density estimator for DOMIAS is not fully fair, since the DOMIAS paper used a more sophisticated density estimator which was found to perform better than the KDE. Of course, the same estimator could also improve the results of Gen-LRA, and PCA+KDE could be computationally cheaper, but these should be checked empirically.

Using PCA may limit the applicability of outlier overfitting detection for outliers with rare categorical values. For example, consider the detection of overfitting on datapoints of French people on the Adult dataset. PCA weights the input dimensions based on how much variance they have, so the indicator for being French would have a very low weight (<1% of the data is French). As a result, the PCA outputs would be very similar between French and non-French people, and Gen-LRA would not be able to detect overfitting affecting French people. Unless I'm completely mistaken about this phenomenon, this should be mentioned as a limitation.

For a similar reason, you should check if datapoints with high DCR score have similarities. It could be that they do, but UMAP is not considering these important. This could change the interpretation of Figure 2 that DCR does not target specific outlier regions.

You should also discuss the fact that Ward et al. (2024) report a very similar finding to your Figure 2 with their MIA. As a part of this, it would be interesting to see analogues of Figure 2 for the other MIAs used as baselines.

Please include separate results from each dataset in addition to the mean results across datasets. The datasets could have significant performance differences that the aggregation hides. I'm also not sure if the standard deviations of performance across different datasets are meaningful in any way.

Minor points:
- The paper should make the differences between DOMIAS and Gen-LRA clearer, since the methods are fairly similar.
- It not clear what $\mathbb{P}\cup \{x^*\}$ precisely is, which makes the motivation leading to Equation 4 seem a bit handwavy.
- Contribution 1: this sentence is a bit unclear, making it seem like the null and alternative hypotheses are the same.
- Line 172: capitalise "equation 4".
- Line 266: missing parenthesis.
- Line 346: "scale" is ambiguous, I would suggest "normalise" if that is what you are doing.
- Several references are missing the publication forum, for example Durkan et al. (2019), Ganev and De Cristofaro (2023).

**Questions:**

No further questions.

---

### Official Review · Reviewer_6TYk · 2024-10-30

**Soundness:** 2
**Presentation:** 3
**Contribution:** 2
**Rating:** 3
**Confidence:** 4

**Summary:**

This paper introduces Gen-LRA, a novel membership inference attack (MIA) methodology for evaluating privacy risks in synthetic tabular data. The authors propose a hypothesis testing framework that computes a likelihood ratio specifically targeted at identifying any local overfitting of the target record. The method requires minimal assumptions, just access to the released synthetic dataset and a reference dataset. They find their method to outperform baselines from the literature across 15 datasets. They further find their method to be particularly successful against outliers, in contrast with other MIAs from the literature.

**Strengths:**

- Technically novel, and interesting, way to compute the membership inference inference signal from synthetic data. The method is theoretically grounded, computationally efficient and relies on limited assumptions for the attacker.
- They show the method to outperform a range of MIAs from the literature
- Comprehensive evaluation of the attack across 15 datasets
- Authors include intuitive examples (eg Fig 1 and Sec 6.2) that are well explained and help the understanding of the paper.

**Weaknesses:**

(More details see questions)

- My main concern comes down to a lack of related work being discussed. A range of important works have studied MIAs against synthetic tabular data using shadow modeling [1,2,3]. While I understand that these works are computationally more expensive and additionally rely on the attacker's knowledge of the training algorithm, I find these works to be very relevant to position this paper and its findings.
- Limited secondary insights with experimental depth. For instance, to make the claim that the method works better for outliers (especially compared to other methods), section 5.3 is mostly anecdotal.

[1] Stadler, T., Oprisanu, B., & Troncoso, C. (2022). Synthetic data–anonymisation groundhog day. In 31st USENIX Security Symposium (USENIX Security 22) (pp. 1451-1468).

[2] Houssiau, F., Jordon, J., Cohen, S. N., Daniel, O., Elliott, A., Geddes, J., ... & Szpruch, L. TAPAS: a Toolbox for Adversarial Privacy Auditing of Synthetic Data. In NeurIPS 2022 Workshop on Synthetic Data for Empowering ML Research.

[3] Meeus, M., Guepin, F., Creţu, A. M., & de Montjoye, Y. A. (2023, September). Achilles’ heels: vulnerable record identification in synthetic data publishing. In European Symposium on Research in Computer Security (pp. 380-399). Cham: Springer Nature Switzerland.

**Questions:**

- Can you expand the related work to also include the shadow-modeling based MIAs?

- To truly understand the contribution, could you implement the shadow-modeling based MIAs [1,2,3] as well and report their results? Right now, the Gen-LRA method seems to be better than the prior work you consider, and does so with limited assumptions for the attacker and with limited computational cost. How does this change when  the attacker now (i) has knowledge of the training algorithm and (ii) has the computational resources to train shadow models? Could authors implement these shadow-model MIAs and report the results alongside Gen-LRA? This would help to position the method and its results in the literature, giving a clear understanding of the impact of certain assumptions and computational cost on the MIA results.

- Similarly, the work on shadow modeling MIAs also discusses disparate vulnerability of outliers [1,2,3]. Stadler et al [1] finds outliers to be more vulnerable than randomly selected records, while Meeus et al [3] proposes a method to identify more vulnerable records. Could authors have more elaborate results for the outlier discussion (e.g. show MIA results for outliers vs random points across datasets) and relate these findings to prior work? While the fact that Gen-LRA focuses on outliers is distinct from distance-based methods, these findings might not be very different than the ones in shadow-modeling based MIAs.

---

### Official Review · Reviewer_Yeu8 · 2024-11-03

**Soundness:** 3
**Presentation:** 3
**Contribution:** 2
**Rating:** 5
**Confidence:** 3

**Summary:**

The paper proposes a new approach to do membership inference attacks for tabular data generative models. The approach first estimates the distributions of (1) the reference samples plus the target sample and (2) the reference samples with kernel density estimation, and then computes the density ratio of synthetic samples over these two distributions. The intuition is that, if the target sample were used in training, the density of synthetic samples on distribution (1) would be higher. Results across various datasets and models show that the proposed approach yields better AUC-ROC and TPR at low FPRs.

**Strengths:**

* The proposed method is simple and effective.

* In general, the writing of the paper is clear.

* The paper has demonstrated results on many datasets and models.

**Weaknesses:**

* The assumption that the reference data is available to the attacker is too strong.

* The title and the abstract do not reflect the scope and constraint of the method sufficiently.

**Questions:**

First, I would like to point out that I am not fully up-to-date on the literature regarding membership inference attacks, especially those involving tabular data. As a result, I may be unable to assess the novelty of this work and might not be familiar with the common settings examined in recent literature.

1. The paper assumes the reference data is available to the attacker. This does not seem to be very realistic to me. Section 1 discussed that a common scenario for synthetic data release is that the data owner wants to release data for open research. This implies that such data is not available to the public before that (if such data is already available, then there is no motivation or value for the data owner to release an additional dataset). That means that the attacker does not have access to the reference data either. The prior work I knew often considered attacks that do not make such assumptions (e.g., https://arxiv.org/pdf/1705.07663 and https://arxiv.org/pdf/1909.03935).

    The paper claims that this setting is realistic in Section 2: "We assume this in practice because this represents a plausible scenario for the owner of S as an attacker may be able to find comparable data in the real world..." Unfortuantely, I do not fully understand this example. It would be great if the author can explain it in more detail in the rebuttal.

2. Continuing on the above point, the paper needs to make it clearer what assumptions each of the baseline methods in Section 5 make. Which of them also makes the assumption that reference data is available to the attacker? This would clarify whether the claimed improvement comes from the relaxation of the assumptions or the fundamental advances of the algorithm itself.

3. The paper only evaluates the proposed algorithm on tabular data. But this is not reflected in the title and abstract. By reading only the title and the abstract, the readers might be misled to think that the paper proposes and evaluates the attack on diverse data types.

    I think it is important to clarify that, as the proposed approach relies on kernel density estimation, which (as discussed in the paper) does not scale well with the data dimension. (The proposed approach relies on dimension-reduction techniques to tackle the issue.) Therefore, it is unclear if such a pipeline can work well on other more high-dimensional and complicated data such as images and text.

4. How do you determine the kernel size and the type of the kernel in the experiments? Is the algorithm sensitive to that?

5. Section 5 mentioned that "For Gen-LRA, we found that the choice of k can have a small impact on the performance of the attack (See Appendix A.3), we therefore use the results of the best k choice for each run as the goal for an MIA is to characterize the maximal empirical privacy risk." I understand that choosing the best k could help "characterize the maximal empirical privacy risk". However, this table is mainly for comparing between different baselines. The comparison would be unfair if you chose the best hyper-parameter for your own approach while not doing that for the baseline methods.

7. The discussion in Section 6.2 is nice, but it would be more self-contained if the paper could describe how DCR works in the main text.


Other minor questions:

1. Section 1: "We demonstrate that Gen-LRA identifies a different source of privacy leakage relative to other commonly used MIAs." It would be better to clarify what "the different source" means here. I could only understand it after reading Section 5.

2. Line 116 and 117: what are M and D? These notations do not seem consistent with what was used before.

3. Line 127: typo on the left quotation mark

4. Line 266: missing a )

**Details Of Ethics Concerns:**

The paper focuses on membership inference attacks, which could be leveraged by adversaries to launch privacy attacks.

---

### Official Review · Reviewer_cCQH · 2024-11-03

**Soundness:** 2
**Presentation:** 3
**Contribution:** 2
**Rating:** 3
**Confidence:** 4

**Summary:**

This paper introduces the Generative Likelihood Ratio Attack (Gen-LRA), a novel membership inference attack specifically aimed at detecting privacy leakage due to overfitting in generative models. Unlike prior methods, Gen-LRA employs a likelihood ratio-based hypothesis testing approach to infer membership without requiring extensive knowledge of the model structure or parameters. By leveraging density estimation techniques, the authors assess whether synthetic data generated by a model is overfitting to specific training data points, particularly in regions with outliers. The authors demonstrate that Gen-LRA significantly outperforms existing MIA methods across various generative architectures and datasets, with particular success in scenarios with low false positive rates, highlighting the nuanced privacy risks associated with generative models.

**Strengths:**

This paper introduces the Generative Likelihood Ratio Attack (Gen-LRA), a novel membership inference attack specifically aimed at detecting privacy leakage due to overfitting in generative models. Unlike prior methods, Gen-LRA employs a likelihood ratio-based hypothesis testing approach to infer membership without requiring extensive knowledge of the model structure or parameters. By leveraging density estimation techniques, the authors assess whether synthetic data generated by a model is overfitting to specific training data points, particularly in regions with outliers. The authors demonstrate that Gen-LRA significantly outperforms existing MIA methods across various generative architectures and datasets, with particular success in scenarios with low false positive rates, highlighting the nuanced privacy risks associated with generative models.

**Weaknesses:**

1. The effectiveness of Gen-LRA depends heavily on accurate density estimation, which can be challenging in high-dimensional data settings. The use of kernel density estimation (KDE) or principal component analysis (PCA) for dimensionality reduction may limit applicability and accuracy. This limitation is critical because the success of the Gen-LRA method hinges on reliable density estimation, which becomes less accurate in high-dimensional spaces without significant computational expense. Inaccuracies here can undermine the method's robustness, making this the most pressing limitation.
2. Although Gen-LRA performs well at low false positive rates, its reliance on outlier detection may lead to elevated false positives in datasets with inherently high variability or complex distributions. False positives can impair the practical applicability of Gen-LRA in privacy-sensitive contexts, as overly cautious results may lead to unnecessary restrictions on data release.
3. Gen-LRA presumes that privacy leakage primarily stems from overfitting, potentially overlooking other forms of leakage that may not manifest as local overfitting. This could lead to incomplete privacy assessments, as the Gen-LRA approach might miss privacy vulnerabilities that do not align with the overfitting model. Expanding Gen-LRA’s scope to address other leakage types could enhance its overall utility.

**Questions:**

1.The manuscript lacks a clear explanation of the practical utility of applying MIA to synthetic data. It remains unclear why synthetic data was chosen as the focus, rather than real-world or other benchmark datasets. The authors are encouraged to provide references in the Related Work section to strengthen the justification for studying synthetic data specifically. Expounding on the unique relevance of synthetic data to MIA would better demonstrate the necessity and contributions of this study.
2.Several typographical errors and repeated references are present in the reference section, such as on Line 527 and Line 729. A thorough review of the references is recommended to ensure accuracy and consistency across all citations.

---

### Official Review · Reviewer_tZB4 · 2024-11-04

**Soundness:** 3
**Presentation:** 3
**Contribution:** 3
**Rating:** 8
**Confidence:** 4

**Summary:**

The paper describes a membership inference attack on generative models. It requires a set of examples generated by the model, S, and a set of reference examples, R, presumably from the same distribution as the data the model was trained on. Then to guess whether some new point x* was part of the training data, it estimates the likelihood ratio of S between a model trained on R vs. a model trained on $R \cup \{x*\}$ using two kernel density estimators. It then thresholds on the likelihood ratio. Experimental results demonstrate impressive improvements compared to baseline models, particularly when evaluated with the critical "true positive rate at low false positive rate" metric.

**Strengths:**

The idea of performing MIA on a generative model by using likelihood ratio of generated data between models with and without the targeted example is very natural and efficient. I'm not surprised that it is very effective, as demonstrated in the experiments. The paper is mostly well-written and well-motivated, and to my knowledge original.

**Weaknesses:**

I'm afraid the specific approach of using kernel density estimators will limit the method's applicability to low-dimensional tabular datasets. I would love to see this idea generalized to higher-dimensional data, probably using something that will scale better than KDEs.

**Questions:**

1. Although I could follow the gist of the idea, some of the notation is not precisely defined. $p_{\mathbb{P} \cup x*}$. It might be clearer to skip Eq.s 3/4 and jump to Eq 5.
1. Do you have any ideas for how to generalize this to forms of data that are not amenable to KDEs (even after applying PCA)?
1. Section 5.3 is not clear to me. What exactly is the experiment here, and what is it supposed to demonstrate?

---

### Note · Authors · 2024-11-27

**Comment:**

We thank the reviewers for their high quality and helpful reviews. Given this feedback, we believe that this work would be better presented with additional experiments and writing revisions that are outside of the scope of this rebuttal window. For these reasons, we are withdrawing our submission.

**Withdrawal Confirmation:**

I have read and agree with the venue's withdrawal policy on behalf of myself and my co-authors.